# Leveraging Multivariable Linear Regression Analysis to Identify Patients with Anterior Cruciate Ligament Deficiency Using a Composite Index of the Knee Flexion and Muscle Force

**DOI:** 10.3390/bioengineering10030284

**Published:** 2023-02-22

**Authors:** Haoran Li, Hongshi Huang, Shuang Ren, Qiguo Rong

**Affiliations:** 1Department of Mechanics and Engineering Science, College of Engineering, Peking University, Beijing 100871, China; 2Department of Sports Medicine, Peking University Third Hospital, Institute of Sports Medicine of Peking University, Beijing 100871, China

**Keywords:** composite index, characteristic points, multivariable linear regression, anterior cruciate ligament deficiency

## Abstract

Patients with anterior cruciate ligament (ACL) deficiency (ACLD) tend to have altered lower extremity kinematics and dynamics. Clinical diagnosis of ACLD requires more objective and convenient evaluation criteria. Twenty-five patients with ACLD before ACL reconstruction and nine healthy volunteers were recruited. Five experimental jogging data sets of each participant were collected and calculated using a musculoskeletal model. The resulting knee flexion and muscle force data were analyzed using a *t*-test for characteristic points, which were the time points in the gait cycle when the most significant difference between the two groups was observed. The data of the characteristic points were processed with principal component analysis to generate a composite index for multivariable linear regression. The accuracy rate of the regression model in diagnosing patients with ACLD was 81.4%. This study demonstrates that the multivariable linear regression model and composite index can be used to diagnose patients with ACLD. The composite index and characteristic points can be clinically objective and can be used to extract effective information quickly and conveniently.

## 1. Introduction

Anterior cruciate ligament (ACL) deficiency (ACLD) is a common injury in people who play sports. The ACL plays an important role in maintaining the stability of the knee joint. However, because of the complexity of the knee joint and ACL, it is difficult to conduct kinematic and dynamic research on patients with ACLD. Studies have focused on building a mechanical model of ACL in vitro [1,2]. Since the establishment of a muscle model by Zajac et al. [3], musculoskeletal models have improved [4,5]. With the help of musculoskeletal models, many studies have investigated the kinematics and dynamics in ACLD-affected knees. Some studies have shown that patients with ACLD adopted quadricep avoidance [6,7] and a stiffening strategy [8], resulting in reductions in the knee flexion moment and peak knee flexion angle. ACLD affects a patient’s gait patterns and further kinematics and dynamics [9]. Ren et al. [10] and Yin et al. [11], respectively, studied the kinematics and dynamics in patients with ACLD. Shelburne et al. [7] considered the role of muscles, explaining that in patients with ACLD, quadricep avoidance occurred to restore anterior tibial translation. Furthermore, increasing hamstring force was also sufficient, implying muscle compensation in the knee instability. Even after ACL reconstruction, patients still have a high risk of osteoarthritis [12,13] because of the loss of normal muscle compensation in patients with ACL reconstruction [14,15].

The clinical diagnosis of ACLD is complicated and expensive, and the diagnosis process requires the subjective judgment of clinicians. For auxiliary diagnosis, many studies have used statistics to study the gait of ACLD. Christian et al. [16] trained the gait trajectory points of patients with ACLD using support vector machines (SVMs) to extract trajectory features. Berruto et al. [17] counted the fluctuation range of the acceleration of the patient’s legs with one ACL reconstruction in a pivot-shift test and demonstrated a significant difference between the ACLD-affected and contralateral sides. Zeng et al. [18] used kinematic data extracted by a motion capture system as features for neural network training. Kokkotics et al. [19] used different machine learning methods to identify patients with ACLD and ACL reconstruction from kinematics and dynamics data. However, only a few studies have used kinematics and dynamics data to diagnose patients with ACLD. There are even fewer studies that can be directly reproduced and can rapidly diagnose patients.

The feature choice is the most important variable, regardless of the statistical method. Reinbolt et al. [20] performed *t*-tests on the entire gait cycle and selected the peak points of the statistics as features to predict outcomes of rectus femoris transfer surgeries. Principal component analysis (PCA) has been widely used for dimensionality reduction and feature extraction [21]. Armstrong et al. [22] used PCA to extract the feature points of kinematics and reconstruct the kinematics process. Based on multiple parameters extracted from gait data, some indexes were developed to identify walking patterns of normal [23] and abnormal [24,25,26,27] gait. Schutte et al. [26] proposed a normalcy index to reflect gait deviations from the mean of normal gait. Liu et al. [28] assessed the abnormal gait in patients with ACLD using the normalcy index calculated by PCA based on kinematics and dynamics data. Similarly, Rozumalski et al. [29] combined a single muscle strength score using PCA to describe the overall lower body joint strength. Hicks et al. [30] used this score as a variable for multivariable regression to study crouch gait. Their regression model had 71% classification accuracy when the parameters were analyzed in detail. However, few studies have combined kinematics and muscle forces to extract features.

This study was performed to identify patients with ACLD using multivariable linear regression through a composite index that combined kinematics and muscle forces.

## 2. Materials and Methods

### 2.1. Participants

Twenty-five patients with unilateral chronic ACLD (the contralateral side was intact) were recruited before ACL reconstruction (ACLD group). Their knees had been injured 6 months to 4 years before testing. Most injuries occurred during basketball. Exclusion criteria were that the patient had no prior ACL and concomitant meniscal and ligament rupture and no history of musculoskeletal disease of the hip or ankle. Their physical activity levels were assessed by the Tegner score, which is a reliable and valid tool for assessing the activity level of patients with ACLD [31]. Average activity level of all patients was normal before knee injuries (score range 3.0–6.0). A control group comprising nine healthy volunteers with no history of musculoskeletal injury or surgery in the lower extremities was selected (Control group). All participants were young males to rule out biomechanical differences between sexes [32]. Ethical approval was obtained from the university’s ethics committee, and written informed consent was obtained from all participants. The morphological data are shown in Table 1, and the participants’ characteristics of the groups were not significantly different.

### 2.2. Data Collection and Modeling Analysis

From January 2014 to December 2016, the experimental 3D data were collected while the patients were jogging using an optical motion capture system (Vicon MX; Oxford Metrics, Yarnton, Oxfordshire, UK). The marker trajectory data were filtered at 12 Hz, and the force data were filtered at 100 Hz using a low-pass Butterworth filter. To track the segmental motion during jogging, all participants had a set of markers attached to their anatomical lower limbs at specific locations based on the plug-in-gait model. The participants were asked to run along a 10-m path at a self-selected speed, and the kinematic data were recorded by eight cameras. No participants reported pain during jogging. Ground reaction forces were collected using two embedded force plates at a sampling rate of 1000 Hz (AMTI, Advanced Mechanical Technology Inc., Watertown, MA, USA). Each participant stepped on the force plates at their self-selected speed. For each participant, five successful jogging trials were recorded, and these results were imported into multi-body dynamics software, AnyBody Modeling System version 6.0.5 (AnyBody Technology, Aalborg, Denmark), to estimate the kinetics of the knee joint.

A lower extremity model [33] implemented in the AnyBody Modeling System was used for the analysis. The model comprised 12 body segments, and 11 joints were used to connect the segments. Six joint degrees of freedom were considered for each leg, with a spherical joint with three degrees of freedom for the hip joint and a universal joint with two degrees of freedom for the ankle joint. The knee joint was modeled as a hinge joint with one degree of freedom because of the soft tissue artifact error [34]. Based on the morphological parameters measured from each subject, each model was scaled with a mass–fat scaling algorithm to perform the subject-specific jogging simulation. The min/max recruitment principle solver based on the optimization of the objective function [35,36], which has good numerical convergence and physiological representation, was used to predict the muscle force during the inverse dynamics analysis. The objective function is generally formulated as follows [5]:(1)$$Minimize\;max\left(\frac{f_{i}^{\left(M\right)}}{N_{i}}\right)$$

Subject to
(2)$$Cf=d,\;\;0\leq f_{i}^{\left(M\right)}\leq N_{i}\;\;,\;\;i\in \left\{1,\cdots ,n^{\left(M\right)}\right\}$$
where $n^{\left(M\right)}$ is the number of muscles, $f_{i}^{\left(M\right)}$ is the respective muscle force, and $N_{i}$ is the strength of the muscle. f contains all unknown forces in the optimization problem. C is the coefficient-matrix for the unknown forces. d contains all known applied loads and inertia forces. Muscle parameters were obtained from a comprehensive musculoskeletal geometry dataset [37]. Some studies have validated the ability of computational muscle forces [38,39].

### 2.3. Muscle Data Processing

According to the characteristics of the model and anatomy related to the knee muscles, force data were output from 13 muscles: rectus femoris (RF), popliteus (POP), vastus (VAS), gastrocnemius lateralis (GL), gastrocnemius medialis (GM), soleus medialis (SOLm), soleus lateralis (SOLl), semitendinosus (ST), semimembranosus (SM), proximal sartorius (SAp), distal sartorius (SAd), biceps femoris long head (BFlh), and biceps femoris short head (BFsh). For each participant, separate simulations were performed based on the data from five different jogging trials. The average values of the five calculations were used to perform dynamics analysis using MATLAB version 2019b (MathWorks, Natick, MA, USA).

The acquired muscle force data was processed in a dimensionless manner, and the nondimensionalization of the force data was divided by the subject’s gravity (mass × 9.8) [40]. To investigate one gait cycle, the kinematics and dynamics data were interpolated to a 0%–100% gait cycle. Additionally, to more intuitively study the patterns of knee flexion and muscle strength, all flexion and muscle data were normalized to their maximal muscle force within that cycle, leading to a normalized amplitude between 0 and 1 [41].

### 2.4. Extracting Features

In this study, PCA was adopted as a statistical method for data dimensionality reduction. The main algorithm of PCA is to map the original n-dimensional data to a new k-dimensional feature that retains the largest variance. Thus, the other parts where the variance is close to zero can be ignored and the loss of information is guaranteed to be small. The flow of PCA is as follows:Collect an m × n matrix *G*, where m is the sample size and n is the n-dimensional variable.Subtract the respective mean from each variable.Compute the covariance matrix of the de-averaged matrix.Calculate the eigenvalues and eigenvectors of the covariance matrix by singular value decomposition.Sort the eigenvalues from large to small, and select the largest k eigenvalues among them. In this study, the ratio of selected eigenvalues to the sum of all eigenvalues was used to assess the information content. Arrange the eigenvectors in the same order as the eigenvalues to form a matrix of principal component coefficients (PCcoeff).Transform the data into a new space, i.e., the new data samples = *G* × PCcoeff. The first k columns are the required features.

Therefore, data are reduced to k dimensions. If a new sample (1 × n) needs to be predicted, perform the same de-average operation first (the centered sample = the new sample—the variable mean of *G* from Step 2). Then, the centered sample × PCcoeff produces the predicted value of the sample after the same processing, and the first k columns can be selected as the features of the new sample.

Inspired by the NI index [28] and the strength score [29,30], the first three features used in the calculation are the average force of each muscle during the stance/swing phase of each participant and the value of each person’s knee flexion during the swing phase. The specific process was to first obtain the average value of the stance/swing phase of 13 muscle forces and then perform PCA on the average muscle force to obtain a column variable. PCA was also used to process the knee flexion data in the swing phase to obtain a column variable. Only one column variable for each feature after PCA was used because the information content was sufficiently large. For the data of knee flexion in the stance phase and other columns after PCA, their regression parameters in the following multivariable regression were not significant and did not affect the final accuracy.

### 2.5. Composite Index

In addition to the above features, a composite index containing the data of the knee flexion and muscle forces’ characteristic points, which were the time points in the gait cycle when the most significant difference was observed between the two groups, was used in this study. Comparing the ACLD and Control groups based on knee flexion and muscle force data for all participants, *t*-tests were performed at each point during a 0–100% gait cycle. The data of the characteristic points (*p* < 0.05 and *p* values were minimal) were finally filtered out as a matrix to calculate the composite index. The selection method of the characteristic points is shown in Figure 1 (using the rectus femoris as an example), and the filtered characteristic points are shown in Figure 2. The muscle force/knee flexion at these characteristic points were filtered out to form a matrix, where the rows of the matrix were the number of participants and the columns of the matrix were the number of characteristic points. Finally, PCA was used to process this matrix to select the representative columns as features.

### 2.6. Statistical Analysis

Using the above variables and samples, we built a multivariable linear regression model. The general form of this model is:(3)$$Y_{diagnosis}=\beta_{0}+\beta_{1}X_{1}+\beta_{2}X_{2}+\cdots +\beta_{q}X_{q}$$

The outcome variable $Y_{diagnosis}$ was a diagnosis of whether the participant was a patient with ACLD, such that positive values correspond with patients with ACLD and negative values correspond to the Control group. In the data used for training, the ACLD group had $Y_{diagnosis}=1$ and the Control group had $Y_{diagnosis}=-1$. $X_{i}$ are the predictive features in the data obtained above and $\beta_{i}$ are the linear weighting coefficients for the predictive features. The formulation and prediction of the model were conducted in MATLAB.

## 3. Results

For these 43 samples (the affected legs of the 25 patients in the ACLD group and both legs of the 9 participants in the Control group), the number of variables selected will affect the final accuracy. The final prediction accuracy changes with feature selection changes are shown in Table 2, where 5-fold cross-validation was used to estimate the predictive ability of the regression model. The second column showed whether only the composite index was used as features. If not, the first column showed the first three features (knee flexion and mean values of muscle force during the stance/swing phase) + the number of features retained in the composite index. If yes, the first column showed only the number of features retained in the composite index. The composite index produced eight features when 90% of the PCA information content was preserved. Therefore, when just using the composite index as features and considering all eight features produced by the composite index, the maximum accuracy achieved was 81.4%. The last column showed the *p* values in the *t*-test for the coefficients of the features of the composite index during regression. The smaller the *p* value, the more significant the corresponding feature. *p* < 0.001 indicated very significant findings and was replaced by 0.001 in Table 2. When using the first 3 features + the composite index, the accuracy gradually increased as the features produced by the composite index increased. When the features produced by the composite index were more than three, the accuracy remained the same and the *p* value of the newly introduced features increased and was not significant. When the features produced by the composite index were equal to five, the *p* value of the last feature was 0.999, indicating that the newly introduced feature had no new information. For a comprehensive comparison, the optimal condition was to select six features (the first three features + three composite index features), and the accuracy rate after 5-fold cross-validation was 81.4%. For comparison and validation, under the condition of using only three composite index features, the accuracy was 79.1%.

The classification ability evaluation of the optimal condition is shown in Table 3. The actual results of classification and the accuracy, precision, recall, specificity, and F1-score were used to evaluate the classification ability of the regression model under the optimal condition. Most of the actual results were correctly classified. All evaluation criteria were above 80%, which proved the good performance of the regression model.

Finally, multivariable linear regression was performed on all samples, and the resulting model is shown in Table 4. In Table 4, the coefficients of the average muscle force during the swing phase, Composite Index 1, and Composite Index 2 were negative and their absolute values were the largest among all coefficients. The *p* value of Composite Index 1 was less than 0.001, the *p* value of Composite Index 2 was 0.006, and the *p* value of Composite Index 3 was 0.208. The overall *p* value of the regression model was less than 0.001.

## 4. Discussion

The multivariable linear regression model using the composite index was able to predict, with 81.4% accuracy, whether participants had ACLD. Under the optimal condition (Table 3), data were well classified, and the evaluation criteria were greater than 80%. Among them, the value of precision was high (87.0%), meaning that the correct proportion of the samples classified as the ACLD group was high. Our model was very capable in diagnosing patients with ACLD. The F1-score was high (83.3%), indicating that our model was effective.

As shown in Table 2, when the composite index was used as features only, the best accuracy of 81.4% was achieved by retaining all eight variables. With only one variable of the composite index, there was still 72.1% accuracy. After importing the first three features and retaining three features of the composite index, the best accuracy of 81.4% was achieved. Therefore, this composite index characterized the information of kinematics and dynamics. Using only the three features of the composite index can also achieve an accuracy of 79.1%. Based on the optimal condition of six features, when more composite index features were imported, the accuracy remained unchanged and the *p* value of the coefficient continued to increase closer to 1, indicating that the introduction of more features was no longer significant. Therefore, the composite index contained more information in the model. Most information in knee flexion and muscle force can be covered in the composite index.

Interpretation of the model must be taken with caution (Table 4). The R-squared value of the regression was 0.542, which indicated that the model was able to explain 54.2% of the variance in the diagnosis of patients with ACLD [30]. For all samples, the optimal features used for regression were significant at the *p* = 0.21 level by the *t*-test. For the F-test on the model, *p* < 0.001 indicated that the fitting process of the model was very significant. The root mean squared error, which estimated the standard deviation of the error distribution, was equal to 0.73, indicating that the model fit well. As shown in Table 4, there was a significant (*p* < 0.001) negative relationship between Composite Index 1 and the diagnostic outcome, with an expected 2.3055 decline in the final outcome for each one-point increase in the first variable of the composite index. Additionally, there was also a significant (*p* < 0.05) negative relationship between Composite Index 2 and the outcome, indicating that the composite index, especially the first two variables, played the most important role in the regression model.

The composite index was determined by the data of the characteristic points in muscle force and knee flexion (Figure 2). Each muscle selected was associated with the knee joint, which aids in understanding gait pathology and planning treatment using gait analysis and biomechanical models [20,30]. As shown in Figure 2, most of the characteristic points in thigh muscles were concentrated at the terminal of the stance phase, which also corresponded to the previous studies, especially decreased quadriceps [6,7] and increased hamstring [7,42,43]. Alternatively, although tibialis triceps were active during the mid-stance phase, they had more of an impact on ankle dorsiflexion during this period and thus were not significantly different in patients with ACLD [44]. In this study, the muscle force and knee flexion were normalized. Therefore, in further research or clinical diagnosis and treatment, even if the muscle force or knee flexion is obtained in different principles, the characteristic points in Figure 2 can still be directly selected.

With further validation, the regression model can be used to aid clinical practice [30]. Table 5 describes the characteristics of two hypothetical subjects from the ACLD and Control groups. These two subjects have feature values close to the mean of their respective groups. Their expected final outcomes are 0.6164 and −0.4672, respectively, which can be clearly classified into the ACLD group and Control group. Of all the features, the expected improvements in Composite Index 1 and Composite Index 2 have the most impact on the final outcomes. Notably, the subject values of the ACLD group are all less than 0, while the subject values of the Control group are all greater than 0. With *t*-tests between the two groups on the composite index, we were able to obtain *p* < 0.001 for the first variable of the composite index and *p* < 0.05 for the second variable, verifying the validity of the composite index in the regression model and demonstrating that using only the composite index is also a successful evaluation index, similar to the normalcy index [28] and strength score [29,30].

Some limitations of this study should be noted. First, some patients with ACLD also had meniscus injuries. One study [45] has shown that about 40% to 80% of patients with ACLD have a concurrent meniscal injury. Grouping the data more deeply will help improve our accuracy. Second, data quality can be further improved. The compensatory patterns in the knee joint change depending on the time after ACLD. Limited by clinical data, the period of patients’ injuries in this study was not concentrated. Unconcentrated data may affect the accuracy of the final results. Third, electromyography (EMG) data can be introduced. EMG can assist in validating the muscle forces obtained from the calculations [46]. In addition, EMG data can be directly involved in the calculation to obtain a composite index. Fourth, the results of the composite index need further validation for the assessment of walking.

## 5. Conclusions

We built a multivariable linear regression model to diagnose patients with ACLD using a composite index that combined knee flexion and muscle forces. This statistical model and composite index can aid clinical diagnosis. The composite index and characteristic points can help avoid complex subjective diagnosis in clinical practice and can be used to extract effective information more quickly and conveniently for diagnosis.

## Figures and Tables

**Figure 1 bioengineering-10-00284-f001:**
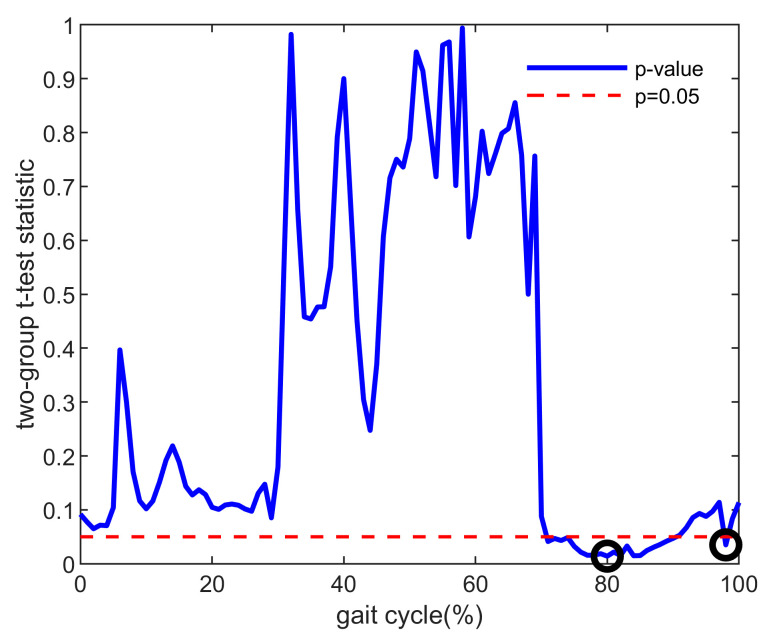
Example of *p* values between the ACLD and Control groups (taking the muscle force of rectus femoris as an example). The red dotted line means that the *p* value equals 0.05. The black circle is the selected characteristic point used to calculate the composite index, which is the minimum value in the range of significant differences (*p* < 0.05). ACLD, anterior cruciate ligament deficiency.

**Figure 2 bioengineering-10-00284-f002:**
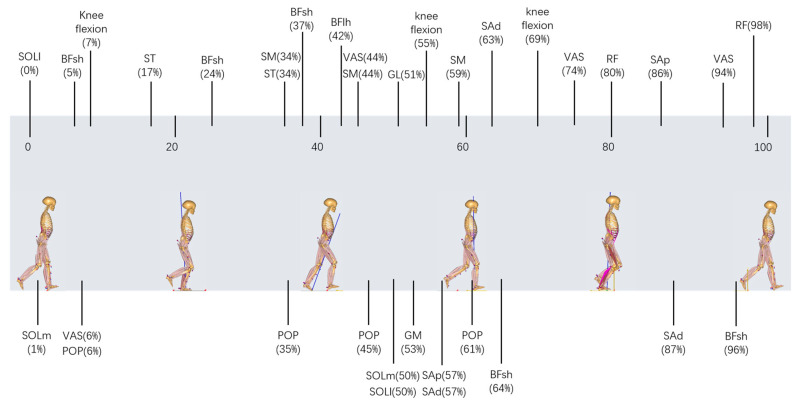
The filtered characteristic points, which were the time points in the gait cycle when the most significant difference (*p* < 0.05 and *p* values were minimal) occurred between the ACLD and Control groups according to the two-sample *t*-test method. Characteristic points are shown at their corresponding points during the 0–100% gait cycle. ACLD, anterior cruciate ligament deficiency.

**Table 1 bioengineering-10-00284-t001:** Characteristics of the participants in each group.

Parameters	Control	ACLD
Age (years)	29.22 ± 5.61	27.20 ± 4.19
Height (cm)	173.67 ± 1.95	178.36 ± 7.23
Weight (kg)	74.06 ± 4.73	82.04 ± 11.55
BMI (kg/m^2^)	24.56 ± 1.62	25.75 ± 2.93
Pace (m/s)	2.32 ± 0.17	2.36 ± 0.25
Time since injury (months)	/	11.10 ± 6.87
Tegner score	/	4.16 ± 1.72

Data are presented as mean ± standard deviation. ACLD, anterior cruciate ligament deficiency; BMI, body mass index.

**Table 2 bioengineering-10-00284-t002:** Accuracy changes as the number of features change.

Number of Features	Use of the Composite Index Only	Accuracy	*p* Value for the *t*-Test of Regression Coefficients for the Composite Index
3 + 1	No	67.4%	0.001
3 + 2	No	74.4%	0.001 0.001
3 + 3	No	81.4%	0.001 0.006 0.208
3 + 4	No	81.4%	0.001 0.008 0.209 0.686
3 + 5	No	81.4%	0.001 0.009 0.225 0.692 0.999
0 + 1	Yes	72.1%	0.001
0 + 3	Yes	79.1%	0.001 0.001 0.75
0 + 8	Yes	81.4%	0.001 0.001 0.74 0.66 0.45 0.01 0.3 0.48

If the composite index is used as features only, the first column shows 0 + the number of features retained in the composite index. If not, the first column shows three features (knee flexion and mean values of muscle force during the stance/swing phase) + the number of features retained in the composite index. *p* < 0.001 indicates very significant results and is replaced by 0.001 in the last column.

**Table 3 bioengineering-10-00284-t003:** Evaluation criteria for the classification ability.

TP	FP	FN	TN	Accuracy	Precision	Recall	Specificity	F1-Score
20	3	5	15	81.4%	87.0%	80.0%	83.3%	83.3%

TP = true positive, samples are classified as positive (ACLD group) and the judgment is correct. FN = false negative, samples are classified as negative (Control group) and the judgment is wrong. FP = false positive, samples are classified as positive and the judgment is wrong. TN = true negative, samples are classified as negative and the judgment is correct. Accuracy = (TP + TN)/(TP + TN + FP + FN). Precision = TP/(TP + FP). Recall = TP/(TP + FN). F1-score represents the harmonic mean of precision and recall. ACLD, anterior cruciate ligament deficiency.

**Table 4 bioengineering-10-00284-t004:** Multivariable linear regression model of all samples.

Features	Coefficient	Standard Error of the Coefficients	*p* Value ^a^
Constant	0.1628	0.1113	0.152
Mean muscle force during the stance phase	0.2384	0.1850	0.205
Average muscle force during the swing phase	−2.5339	0.1968	0.206
Knee flexion during the swing phase	0.7346	0.5518	0.191
Composite Index 1	−2.3055	0.5040	<0.001
Composite Index 2	−1.5697	0.5468	0.006
Composite Index 3	1. 2417	0.9698	0.208
RMSE	0.73		
R-squared	0.542		
*p* value ^b^	<0.001		

^a^ *p* value for the *t*-test of each regression coefficient. ^b^ *p* value for the F-test on the model. Composite index 1–3 represent the first three features of the composite index, respectively. RMSE, root mean squared error.

**Table 5 bioengineering-10-00284-t005:** Characteristics of two hypothetical subjects from the ACLD and Control groups.

Features	ACLD Group		Control Group	
	Subject Value	Expected Improvement	Subject Value	Expected Improvement
Constant	1	0.1628	1	0.1628
Mean muscle force during the stance phase	−0.1757	−0.0419	0.2440	0.0582
Average muscle force during the swing phase	−0.0063	0.0160	0.0088	−0.0223
Knee flexion during the swing phase	−0.0051	−0.0037	0.0071	0.0052
Composite Index 1	−0.1490	0.3435	0.2069	−0.4771
Composite Index 2	−0.0937	0.1470	0.1301	−0.2042
Composite Index 3	−0.0059	−0.0073	0.0082	0.0102
Expected outcome		0.6164		−0.4672

Composite Index 1–3 are respectively the first three features of the composite index. ACLD, anterior cruciate ligament deficiency.

## Data Availability

Not applicable.

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
