# Peer review of "Leveraging Multivariable Linear Regression Analysis to Identify Patients with Anterior Cruciate Ligament Deficiency Using a Composite Index of the Knee Flexion and Muscle Force"

_bioengineering, 2023, doi:10.3390/bioengineering10030284_

Round 1

Reviewer 1 Report

Paper on detecting mobility problems associated with knee joint injuries.

The problem is clearly defined as well as the methodology followed by the authors.

The text is well written and the interpretation of the results seems adequate.

In my opinion, the paper should not be published as it is, as there are still some issues that need clarification.

The measurement protocol does not seem clear enough. Although it is mentioned in the text that each element was tested running, I think the authors mean walking.

otherwise

-If the measurements were taken while running, why not take them on an instrumented treadmill?

How was ensured the uniformity in the paths during measurements ?

How was the passage of individuals over the two force platforms ensured?

Minor corrections in the text:

Pag 3. Line 118, should be massX9,8 (weight is a force!)

Pag 5 line 182 the equation should be numbered.

Reviewer 2 Report

This study evaluated 25 patients with ACL deficiency and nine healthy volunteers using a musculoskeletal model during running. Results from knee flexion and muscle force were analyzed to diagnose ACLD with an accuracy rate of 81.4%. In addition, a multivariable linear regression model and composite index were developed to provide quick and convenient objective evaluation criteria for the clinical diagnosis of ACLD. The paper is well-written and organized, and the authors are commended for their efforts. However, the following minor points may improve the quality of this manuscript and should be addressed properly.

1) L113: this statement is not clear "Five different running trials were simulated based on the experimental data, and the average values were used to perform dynamics analysis using MATLAB version 2019b (MathWorks, Natick, MA, USA)."

2) L116: what about the dimensionless manner of acquired flexion?

3) L116-121: it is unclear here why the authors "dimensionless and normalized the parameters."

4) Please replace the X in the tables with the name of the real variable.

5) How about the validation of the predicted muscle forces?

Reviewer 3 Report

This study is interesting with potential clinical application, but some points should be improved during revision. 

  1. The rationale in the introduction should be strengthen. More studies related to the current aim should be included. The novelity of this study should be emphesize - what it add to curent knowledge. 
  2. The study groups should be better described, ithe inclusion/exclusion criteria for both groups need clarification. What was the physical activity level of subjects? Were they professional athletes before ACL rupture?  The time from ACL injury ranged from 6 months to 4 years. This point is the main problem in data quality, because 6 months after ACL injury it is still the rehabilitation period  with significant deficits in knee function (muscle strength, proprioception, functional performance). On the other hand 4 years after injury without ACL reconstrction a lot of compensatorry patterns may occur. Bot situation significantly influence gait pattern - what was the main outcome measure in this study. I am afrain dthat existance of confounding factors in experimental group may diminish the quality of results. 
  3. The composite index is not described in the methods - It should be presented as separate part of the metods. All steps of composite index clculation should be presented in details. 
  4. The results are presented chaotically and should be deeply revised. The Tables do not have any comments in the text .  The results needs clarification and presentation at more redable way. 
  5. The study limitations are lacking
  6.  I am not  sure that the data obtained are valid and conclusive due to confounding factors related to groups heterogenity (see poitt 2) 

Reviewer 4 Report

Diagnosis of a patient based on performed quantitative tests is not easy. The reason for this is a lot of different information that accurately describes all elements of the motion structure. Therefore, it becomes necessary to search for methods that will support doctors and physiotherapists in identifying various pathological changes, and most importantly - the diagnosis will be quantifiable and measurable.
The method proposed by the authors is a very interesting combination of the use of a biomechanical model of the human motor system with statistical techniques (PCA). An analysis of various variants of the use of 'Composite Index' was performed and the obtained quality of results was indicated.
I very positively perceive the indication of limitations that may affect the interpretation of the model's results.

There is no need to make any corrections to the content of the article.

I have only one question - why do you choose running and not walking for the analysis? Since you're pointing out a potential possible clinical application, you have to consider that no doctor will be evaluating a patient with an ACL disorder while running. Can the indicated results be directly applied to the assessment while walking? If so - please write about it, which will be better received in the context of clinical use.

Round 2

Reviewer 3 Report

Authors have improved the manuscript according to my suggestions. I recommend accept it.